# The Use of Zidovudine Pharmacophore in Multi-Target-Directed Ligands for AIDS Therapy

**DOI:** 10.3390/molecules27238502

**Published:** 2022-12-03

**Authors:** Maria da Conceição Avelino Dias Bianco, Debora Inacio Leite, Frederico Silva Castelo Branco, Nubia Boechat, Elisa Uliassi, Maria Laura Bolognesi, Monica Macedo Bastos

**Affiliations:** 1Programa de Pos-Graduação em Farmacologia e Química Medicinal, Instituto de Ciências Biomédicas—ICB, Universidade Federal do Rio de Janeiro—UFRJ, Av. Carlos Chagas Filho, 373, Bloco L, Rio de Janeiro 21941-902, Brazil; 2Laboratorio de Sintese de Farmacos—LASFAR—Fundacao Oswaldo Cruz, Instituto de Tecnologia em Farmacos—Farmanguinhos—Fiocruz, Rua Sizenando Nabuco, 100—Manguinhos, Rio de Janeiro 21041-000, Brazil; 3Department of Pharmacy and Biotechnology, Alma Mater Studiorum—University of Bologna, Via Belmeloro 6, I-40126 Bologna, Italy

**Keywords:** HIV, AIDS, zidovudine, polypharmacology, multi-target-directed ligand, hybrid, co-drug

## Abstract

The concept of polypharmacology embraces multiple drugs combined in a therapeutic regimen (drug combination or cocktail), fixed dose combinations (FDCs), and a single drug that binds to different targets (multi-target drug). A polypharmacology approach is widely applied in the treatment of acquired immunodeficiency syndrome (AIDS), providing life-saving therapies for millions of people living with HIV. Despite the success in viral load suppression and patient survival of combined antiretroviral therapy (cART), the development of new drugs has become imperative, owing to the emergence of resistant strains and poor adherence to cART. 3′-azido-2′,3′-dideoxythymidine, also known as azidothymidine or zidovudine (AZT), is a widely applied starting scaffold in the search for new compounds, due to its good antiretroviral activity. Through the medicinal chemistry tool of molecular hybridization, AZT has been included in the structure of several compounds allowing for the development of multi-target-directed ligands (MTDLs) as antiretrovirals. This review aims to systematically explore and critically discuss AZT-based compounds as potential MTDLs for the treatment of AIDS. The review findings allowed us to conclude that: (i) AZT hybrids are still worth exploring, as they may provide highly active compounds targeting different steps of the HIV-1 replication cycle; (ii) AZT is a good starting point for the preparation of co-drugs with enhanced cell permeability.

## 1. Introduction

Nowadays, drugs that act on multiple targets seem an increasingly feasible and attractive polypharmacology option for drug discovery [1]. The ‘one gene, one target, one drug’ paradigm has inspired the development of potent and selective ligands for many years in the past. However, there is a general awareness that they are inadequate to deal with the complexity of major chronic diseases or the problem of drug resistance. As most of the currently incurable diseases are caused by multiple factors, the concomitant modulation of several targets through polypharmacology is required [2,3].

The concept of polypharmacology includes three options [4]: multiple drugs that bind to different targets (drug combinations or drug cocktails), fixed dose combinations (FDCs), and single drugs that bind to different targets (multi-target drugs). The effectiveness of the therapeutic approach based on drug cocktails may be compromised by poor patient compliance and the risk of drug–drug interactions, commonly caused by induction or inhibition of hepatic metabolism [5]. However, it is the mainstay for the treatment of several multifactorial diseases, such as Alzheimer’s disease, cancer, respiratory diseases, and acquired immunodeficiency syndrome (AIDS), among others. FDCs improve patient adherence to therapy by bringing together two or more active pharmaceutical ingredients (APIs) in a single formulation. On the other hand, FDC limits dose flexibility and is more expensive than generic drugs. Regarding multi-target drugs, in 2008 Bolognesi and colleagues, based on the potential of polypharmacology in the treatment of neurodegenerative diseases, proposed the term multi-target-directed ligands (MTDLs) to refer to this class of compounds [6,7]. The purpose of this was to better highlight “their ability to interact with the multiple targets thought to be responsible for the disease pathogenesis” and to clearly differentiate them from so-called “promiscuous drugs” [1]. On this basis, in this review the term MTDLs will be used to refer to these compounds (Figure 1).

As mentioned, the polypharmacology approach is currently applied in the treatment of AIDS, which remains a worldwide public health problem. Currently, there is not a cure or an effective vaccine for AIDS; however, the available therapy provides a better quality of life for patients, making it a manageable chronic disease. Treatment consists of combined antiretroviral therapy (cART) [8], and this drug combination strategy is showing good results in the control of the disease. In the case of AIDS, the reduction in viral load to undetectable levels in the bodies of patients is fundamental to prevent disease spread [9].

AIDS is caused by the human immunodeficiency virus type 1 (HIV-1) and type 2 (HIV-2), which belong to the *Retroviridae* family and the *Lentivirus* genus. According to the World Health Organization (WHO) estimation, there were approximately 37.7 million people worldwide living with HIV/AIDS in 2020 [10].

The HIV replication cycle presents several events exclusively related to viral components, which can be targets for chemotherapeutic intervention. Figure 2 illustrates that the HIV life cycle starts when (1) HIV fuses with the surface of a CD4^+^ T cell, and (2) a capsid consisting of the virus’s genome and proteins moves into the cell. (3) The disruption of the capsid shell allows HIV reverse transcriptase to transcribe the viral RNA into DNA. Then, (4) HIV DNA is transported across the nucleus, where the HIV integrase inserts the HIV DNA into the CD4^+^ cell DNA. (5) The host transcription machinery transcribes HIV DNA into new RNA copies, which will be used for the genome of a new virus and to make new HIV proteins. (6) The new HIV RNA and proteins move to the surface of the cell, where a new, immature HIV forms. Finally, (7) the virus is released from the cell, and HIV proteases cleave newly synthesized proteins to create a mature infectious virus. The available drugs act at different stages of this HIV replication cycle and through different mechanisms of action [11]: (1) acting on glycoproteins (gp120 and gp41) or receptors and co-receptors present on the host cell surface (CD4 and CCR5 or CXCR4); (2) by blocking the fixation or fusion process, collectively called “entry inhibitors”; (3) as inhibitors of the reverse transcriptase (RT) enzyme (RTIs), either nucleosides (NRTIs) or non-nucleoside (NNRTIs); (4) as inhibitors of integrase, the enzyme that promotes active integration of the viral DNA double-strand into the host cell genome (integrase inhibitors); or (7) by “competitive” inhibition of proteases (protease inhibitors, PIs). However, drug resistance may arise from each of these drug classes. There are essentially two mechanisms by which resistance to NRTIs can occur: (i) mutations of residues at or near the drug-binding site, and (ii) mutation of the residues that results in reduced incorporation/enhanced removal of the drug into/from its binding site. For NNRTI and PI drug classes, resistance occurs primarily as a result of amino acid mutations within or proximal to the drug-binding site. Drug combinations have been shown to slow down the evolution of resistance, as the simultaneous administration of two drugs, which operate with different mechanisms of action, can reduce the probability of mutations. Consequently, clinical efficacy has been achieved with the introduction of cART therapeutic regimens that include three or more drugs from at least two different drug classes; however, the initial euphoria of the therapeutic advance was quickly dashed by the appearance of strains resistant to different combinations of the available drugs. These factors have called for a renewed drug discovery effort in this area, aiming to obtain new molecules with known or innovative mechanisms of action [11].

Today it is widely recognized that antiretroviral therapy still needs to be improved. All available drugs have important side effects, and the patient often does not adhere to the treatment consistently, resulting in many cases of drug resistance. Drug-resistant HIV occurs when the virus replicates in subtherapeutic concentrations of antiretrovirals and compromises the effectiveness of treatment, so new strategies are necessary [12,13]. Medicinal chemists should concentrate their efforts on the search for new compounds able to overcome all drawbacks of the current therapy and, ultimately, obtain more effective alternatives for the treatment of AIDS.

A widely applied drug in the search for new compounds with antiretroviral activity is 3′-azido-2′,3′- dideoxythymidine, also known as azidothymidine or zidovudine (AZT), an analog of thymidine (Figure 3a). This drug was initially developed as an anti-cancer agent in the 1960s, but at the end of 1980s the United State regulatory agency, the Food and Drug Administration (FDA), approved it as an anti-HIV therapeutic agent. Therefore, it was also the first example of drug repositioning in the history of medicinal chemistry [14,15]. Indicated for adults and children, its administration must occur in combination with other antiretroviral drugs. Nowadays, this drug is commonly used in the prevention of perinatal HIV-1 transmission (vertical transmission) that consists of the use of this drug by the mother before and during delivery, and treatment of the newborn [16]. AZT belongs to the NRTI class and is a chemically modified prodrug, being an analog of thymidine that differs by the presence of an azido group replacing the hydroxyl at the C-3′ position in the deoxyribose ring [17]. The active form of this prodrug is formed upon triphosphorylation at the 5′-OH position in the cell by kinases. The mechanism of action is characterized by the conversion into the corresponding 5′-O-triphosphate that inhibits the virus replication through a competitive binding to RT with subsequent termination of the growing viral DNA strand (Figure 3b). If the copied DNA is not correctly formed, the viral RNA genome can be destroyed by enzymes [18].

Interestingly, some studies show that AZT can partially reverse HIV-associated neurological disorders, such as dementia and peripheral neuropathy, in some patients with advanced infection. However, these effects seem to be limited and diminish with the prolongation of therapy. Furthermore, AZT can also be used to treat other diseases, such as in blocking T-leukemia virus-1 (human lymphoma) and other mammalian retroviruses [19].

A disadvantage in the use of AZT, as well as of other NRTIs, is toxicity. In the case of this drug, the main adverse effects observed are nausea/vomiting, diarrhea, headache, and bone marrow suppression. In order to decrease toxicity, increase potency/selectivity, and optimize the pharmacokinetic profile, medicinal chemists have been interested in modifying AZT structure and obtaining improved treatment options [18,20,21,22].

A recent trend considers AZT as a suitable starting point for new multi-target drug discovery endeavors. This article will cover some of these medicinal chemistry strategies that are aimed at identifying new analogs of AZT with a polypharmacological antiretroviral profile.

## 2. Harnessing AZT for Polypharmacology: Drug Cocktails, FDCs, and MTDLs

As anticipated, both drug cocktails and FDCs are extensively employed as polypharmacological therapies for the treatment of AIDS. Nowadays, there are more than twenty FDCs FDA-approved, and among them, Combivir^®^ (lamivudine + zidovudine) and Trizivir^®^ (lamivudine + zidovudine + abacavir) are those containing AZT together with other NRTIs (Figure 4). Currently, guidelines suggest cART with two NRTIs in combination with a third active drug, the latter consisting of an integrase inhibitor. Thus, the preferred therapy for adults consists of an FDC containing tenofovir + lamivudine + dolutegravir (TLD) [23,24] (Figure 4). In 2021, the FDA approved Cabenuva^®^ (an FDC of the integrase inhibitor, cabotegravir and the NNRTI, rilpivirine, co-packaged for intramuscular use) as a once-monthly or every-two-months treatment.

Clearly, both approaches have advantages and disadvantages, so when determining the treatment of a patient, it is necessary to analyze which would be more appropriate. The main advantage of using drug cocktails is the potential to adjust the dose of each drug according to the patient’s need, providing personalized medicine. Although this cannot be achieved with an FDC, the latter generally has better patient compliance due to the reduction in pharmaceutical forms to be administered [25]. On the other hand, the identification and validation of new drug cocktails are difficult to implement due to the variety of options in terms of the number of pairs and the dosage combinations. As a general feature, associations are more likely to have drug interactions and toxic effects than single drugs [26].

A newer option in drug development for multifactorial diseases or those requiring drug combinations is the search for single-molecule MTDLs. By modulating different targets simultaneously, MTDLs can provide essential advantages over cocktails and FDCs. Compared to cocktails, MTDLs would eliminate the drug interaction problem in addition to facilitating adherence to therapy, since the number of administered pills will be reduced. Compared to FDCs, they have more predictable pharmacokinetics, do not present risk of drug interactions in formulation development, and may be better tolerated by patients [27,28]. MTDLs can be divided into two classes: co-drugs and hybrids (Figure 5) [7]. Co-drugs consist of two distinct synergistic drugs that are connected by a labile covalent bond, where each one acts as a carrier for the other in a mutual manner. These are prodrugs, since they only produce their pharmacological effects after metabolic activation. Conversely, hybrids may consist of two or more pharmacophores with proven activity and/or toxicity, combined in a single compound [28]. The classification of hybrid molecules can be based on their chemical structure or related to their targets. According to the structural classification, hybrids can be called linked, fused, or merged hybrids. Linked hybrids are obtained from two pharmacophoric frameworks connected by a metabolically stable spacer, which is absent in both parent ligands. In fused and merged hybrids no linker is present, but they differ from each other by their degrees of framework integration (Figure 5). For the fused ones, the two active units are connected almost directly, whereas in merged hybrids molecular frameworks overlap through a common structural element. Clearly, merged hybrids have simpler chemical structures and lower molecular weight when compared to the combination of the two precursor compounds [2,4].

Regarding the classification based on targets, according to Barreiro and Fraga [2], two targets belonging to the same biochemical pathways are classified as dual or mixed ligands. When the selected targets are of different biochemical pathways but related to the same pathophysiology, they are classified as symbiotic ligands [2].

A broad theoretical basis is required for the rational design of an MTDL. Classical concepts of medicinal chemistry are used, such as molecular hybridization, which is one of the most important tools [27].

In this review, we provide an overview of MTDLs with reported anti-HIV activity, structurally derived from AZT, as possible future trends in the treatment of AIDS. We decided to explore the literature by selecting the most relevant papers that showed the importance of the AZT scaffold in the polypharmacological profile and structure–activity relationship (SAR) data.

## 3. Linked Hybrids as Potential HIV-1 Replication Inhibitors

Caramasa’s group was the first to develop AZT-based hybrids [29]. By combining the N-3 position of AZT’s thymine ring and NNRTIs, i.e., TSAO-T ([2′,5′-bis-O-(tert-butyldimethylsilyl)-beta-D-ribofuranosyl]thymine]-3′-spiro-5″-(4″-amino-1″,2″-oxathiole-2″,2″-dioxide) and HEPT ([(2-hydroxyethoxy)methyl]-6-(phenylthio)thymine), via a non-cleavable methylene linker of different lengths (n = 3–9), two series of hybrids with the general formula [AZT]-(CH_2_)_n_-[TSAO-T analogs] and [AZT]-(CH_2_)_n_-[HEPT analogs] have been designed and synthesized. Regarding the anti-HIV profile, the most potent hybrid **1** (Figure 6) belongs to the [AZT]-(CH_2_)_n_-[TSAO-T analogs] set and showed a half-maximal effective concentration (EC_50_) of 0.10 µM against HIV-1 in CEM/0 cells, while being not cytotoxic up to 100 µM. Of note, the antiviral activity within this series demonstrated a clear trend in decreasing potency with the increasing number of methylene units of the linker (n > 6). On the other hand, no derivatives of the [AZT]-(CH_2_)_n_-[HEPT analogs] series proved to be active against HIV-1. Both hybrid sets showed no anti-HIV-2 activity in CEM/0 and CEM/TK cells. However, although **1** exerted good anti-HIV-1 effect, it was found to be less potent than the parent compound (Figure 6).

In a follow-up work [30], the authors designed a second series based on AZT and TSAO-T scaffolds aimed at exploring the influence on the antiviral activity of chemically diverse spacers and sugar moieties. The resulting hybrids **2**–**8** (Figure 7), featuring flexible, rigid and polar linkers, displayed comparable anti-HIV-1 activity with that of the prototype **1**. Notably, replacement of AZT in the prototype **1** by 2′,3′-didehydro-2′,3′-dideoxythymidine led to compound **7**, which resulted in a 5- to 10-fold higher inhibitory effect against HIV-1 in CEM/0 and MT-4 cells, respectively, in comparison to **1** (Figure 7). Considering that [AZT]-(CH_2_)_n_-[TSAO-T analogs] might not be recognized by cellular kinases and thus transformed in their active forms, the authors synthesized compounds with a monophosphate at the C-5′-position of the sugar ring. However, no improvements in anti-HIV-1 activity were detected. Interestingly, the phenyl phosphoramidate derivative **6** (Figure 7) showed anti-HIV-2 activity (EC_50_ of 15 ± 7.1 µM).

Pontikis and co-workers designed, synthesized, and biologically evaluated a series of linked hybrids carrying an NRTI and an NNRTI [31]. Hybrids **9**–**11** (Figure 8), obtained by the combination of AZT and a HEPT analog, exhibited the best anti-HIV profile, with EC_50_ values of 2.8, 3.3, and 1.7 µM, respectively. However, **9**–**11** were less active than AZT (EC_50_ = 0.002 µM). In addition, **9**–**11** showed no effects on HIV-2 replication or the HIV-1 resistant strain with the Y181C mutation. Thus, no synergistic effects were observed, perhaps because the linker is unable to allow simultaneous binding of the AZT and HEPT motifs at their respective sites [31].

Natural compounds may also inspire the development of anti-HIV hybrids. Betulin is a natural and abundant pentacyclic triterpene that possesses a broad spectrum of biological activities and inhibits different steps of the HIV replication cycle [32,33,34,35]. SAR studies on the pentacyclic triterpene core demonstrated that the mechanism of antiviral activity depends on the modification pattern at C-3 or C-28. Modification at C-3 led to bevirimat (Figure 9), which is able to inhibit HIV-1 maturation [32,33], while modification at C-28 provided LH55, which blocks HIV-1 entry by inhibiting gp-120 [34,35] (Figure 9). For this reason, betulin emerged as a valuable starting point for the development of novel anti-HIV hybrids.

On this basis, and in search of new anti-HIV hybrids, Xiong and co-workers combined betulin derivatives with AZT, leading to fourteen new hybrids [36]. Among them, nine showed potent anti-HIV activity in the submicromolar range. Particularly, **12** and **13** (Figure 10) displayed EC_50_ values against HIV-1-NL4-3-infected MT-4 cells comparable to those of AZT and bevirimat. Importantly, **12** and **13** outperformed both parent compounds in terms of inhibition of mock-infected MT-4 cell growth (IC_50_). A 2,2-dimethylsuccinyl spacer between the C-28 position of the triterpene scaffold and AZT appeared to be the best linker [36].

Previous works indicated that the introduction of a triazolyl group to pentacyclic triterpenoids could result in compounds with good biological activities [37,38,39,40,41,42,43]. Considering that AZT features an azido group that can react with a terminal alkyne to form the 1,2,3- triazole ring [44,45,46,47,48,49], several triazole derivatives have been developed based on AZT and betulinic acid. 1,2,3-triazole moiety not only provides an easy and fast way for linking two frameworks, but it is also a valuable motif for the development of novel anti-HIV-1 agents. In fact, triazole derivatives can act by inhibiting different HIV-1 enzymes, such as RT, integrase, and protease [50]. As an example, compound **14** (Figure 11), featuring a 1,2,3-triazole as a spacer between AZT and the C-28 position of betulinic acid, exhibited an EC_50_ value of 0.10 µM, which corresponds to that of AZT (EC_50_ = 0.10 µM) and is higher than that of bevirimat (EC_50_ = 0.077 µM) [37]. Compound **14** also displayed toxicity comparable to bevirimat (CC_50_ of 11.2 and 13.2 µM, respectively) and greater than AZT.

Wang and co-workers synthesized novel AZT-betulinic/betulonic acid hybrids using a 1,2,3-triazole as a linker between the C-2 of the triterpenoid acid and the 3′-azido group of AZT [51]. However, none of the three hybrids conjugated with AZT (**15**, **16**, and **17**) displayed significant anti-HIV activity (Figure 12).

A successful attempt at clicking AZT into 1,2,3-triazoles carrying a bulky aromatic group at the C-4 or C-5 position provided potent antivirals [52]. SAR studies pointed out that hybrid **18** (Figure 13), substituted at the C-5 position of the 1,2,3-triazole ring, was more potent (83% inhibition of HIV-1 in CEM-SS cells at 10 µM) than the corresponding C-4 substituted compound **19** (33% inhibition of HIV-1 in CEM-SS cells at 10 µM) [52]. However, **18** and **19** showed a reduced anti-viral activity in a single replication cycle WT HIV assay (EC_50_ = 1 µM and 7.2 µM, respectively) compared to AZT (EC_50_ = 0.14 µM). The authors further characterized the antiviral profile of **18** and **19** against a NNRTI-resistant (NNRTIr) HIV strain (Figure 13). Notably, **18** (EC_50_ = 0.6 µM) had 3.5-fold higher efficacy than **19** (EC_50_ = 2.1 µM) against the resistant strain, but 5-fold lower than AZT (EC_50_ = 0.12 µM).

In a subsequent study, the same authors showed that when the naphthyl group is replaced by a tetrazole as for **20** (Figure 13), a significant loss of activity occurred [53]. Additionally, the substitution of the hydroxyl functionality with the silyl group of compound **21** was detrimental to the anti-HIV activity [54].

The substitution at the C-4′ position of the sugar moiety with a 1,2,3-triazole ring (rather than at the C-3′ position) was evaluated and provided derivative **22** (Figure 14). Although **22** had the best anti-HIV profile within the series, it had a moderate anti-HIV-1 activity (18–62% inhibition at 10 µM), indicating that the substitution at C-4 negatively affected the antiviral profile [55].

For the sake of clarity, it should be noted that while **18–22** share a clear hybrid structure, it is debatable whether they can be classified as MTDLs. This is because the second moiety linked to AZT was not deliberately chosen as carrier of a second pharmacological activity.

Olomola and co-workers developed triazole-based anti-HIV hybrids **23** and **24** (Figure 15) by linking a coumarin-based HIV-1 protease inhibitor (PI) and AZT as RT inhibitor. Hybrids **23** and **24** are able to inhibit the selected HIV targets in a similar manner as dual-acting inhibitors with a balanced activity [56].

Our research group has been developing several compounds with potential anti-HIV activity, including hybrids, which bear the AZT core (Figure 16) [57,58,59]. Knowing the importance of maintaining the terminal hydroxyl group (5′-OH) to provide the active compound, we explored the C-3′ position of the sugar ring to generate potential drug candidates.

Based on a molecular hybridization strategy, AZT was combined with isatin via a 1,2,3-triazole ring leading to hybrid **25** (Figure 16). Remarkably, **25** turned out to be 2-fold more potent (IC_50_ = 0.6 µM) than the anti-HIV drug tenofovir (IC_50_ = 1.2 µM) [57]. This result inspired the development of compound **26** [58] (Figure 16), designed to act against HIV-1 and *Mycobacterium tuberculosis* (Mtb), which is a clinically relevant co-infection. Assays in TMZ cells demonstrated the potential for decreasing the HIV-1 infection by 91% (Figure 16). In 2018, we decided to replace the isatin core with that of efavirenz, a RT inhibitor of the NNRTI class. The novel hybrid **27** showed the lowest IC_50_ value (0.9 µM) and the ability to inhibit HIV-1 RT comparably to tenofovir (Figure 16) [59].

In the search for multi-target compounds against HIV–malaria co-infection, Aminake and co-workers [60] synthesized hybrids carrying AZT and dihydroartemisinin (DHA) or chloroquine (CQ), which are effective anti-malaria scaffolds. Taking into consideration only the HIV-inhibitory activity, compound **28**, featuring a protected C-5′ OH-AZT linked to CQ through a succinyl spacer, was the most potent antiviral hybrid with an IC_50_ of 0.9 µM (Figure 17). This was partially expected since additive in vitro anti-HIV effects were observed with AZT-CQ combination. The presence of a protecting group on the hydroxyl function seemed important for anti-HIV effects. Nevertheless, **28** displayed a reduced activity compared to AZT (IC_50_ = 0.04 µM), but greater than that of CQ (IC_50_ = 12.48 µM). Additionally, it showed cytotoxicity in HeLa cells (CC_50_ = 28.65 ± 5.09 µM), but still with a moderate selectivity (selectivity index (SI) > 30) [60].

Another important co-infection in which the multi-target approach might be particularly suitable is HIV–tuberculosis. Senthilkumar and colleagues combined the C-5′ hydroxyl group of AZT with antimycobacterial fluoroquinolones to afford hybrids **29** and **30** (Figure 18) [61]. As reported in Figure 18, compound **29** proved to be the best HIV-1 replication inhibitor in acutely infected C8166 cells (inhibition of syncytium formation, Syn form) with an EC_50_ of 0.00098 μM, being 15-fold more active than the parent drug AZT. Hybrid **29** showed low toxicity and an SI (CC_50_/EC_50_) > 6000. In addition, **29** turned out to be active against HIV-1_IIIB_ replication in MT-4 cell lines with an EC_50_ of 0.0066 μM. In this assay, compound **30** emerged as the most potent HIV replication inhibitor, with an EC_50_ of 0.0012 µM. Moreover, **30** was moderately toxic with a CC_50_ of 34.05 μM against MT-4 cell lines and an SI = 28.37. In spite of the potent antiviral effects, both **29** and **30** had higher toxicity than AZT.

## 4. Co-Drugs as Potential HIV-1 Inhibitors

Generally speaking, the co-drug approach is based on linking two drugs through a labile covalent bond in a single molecule that acts as a prodrug with an improved therapeutic efficiency/PK/toxicity profile [7,62]. In the specific MTDL context, the two starting molecules should be synergistic drugs, which, following metabolic transformation, have the potential to be released in the same target cells and at the same time. This is a peculiar feature of MTDL co-drugs with respect to combinations (two single drugs, each one with an individual pharmacokinetic profile) [63,64]. In 1988, Busso and co-workers employed this strategy, seeking to obtain a superior pharmacological effect with nucleotides possessing a dimeric structure [65]. For this purpose, a series of nucleotide homo- and heterodimers were synthesized and their in vitro antiviral and cytotoxicity properties compared to their parent monomers (**31**–**34**, Figure 19). The authors reported that nucleotide dimers linked via a phosphate bridge have enhanced in vitro anti-HIV potency in comparison with the monomers, as presented in Figure 19. The anti-HIV activity demonstrated by the dimers was quantified through 50% effective dose (ED_50_) values. All compounds were more potent HIV inhibitors when compared with the reference molecules as well as AZT + 2′,3′-dideoxyadenosine (ddA) combination, which exhibited the highest inhibitory activity. The dimer **31** stood out as it demonstrated an ED_50_ of 0.7 µM (Figure 19). According to the value of 50% inhibitory dose (ID_50_), which is indicative of cell viability, **32** showed the highest toxicity with an ID_50_ of 60 µM (Figure 19).

Ijichi and co-workers [66] developed co-drugs and evaluated these new nucleotide heterodimers on HIV-1 replication, including resistant mutants. The novel heterodimers were designed linking one of the dideoxynucleosides (AZT or didanosine (ddI)) and the 6-[3,5-(dimethyl-phenyl)thio]-5-ethyl-1-[(2-hydroxyethoxy)methyl]uracil (E-HEPU-dM) or ribavirin. The in vitro inhibitory effects on HIV-1 replication revealed that all compounds featuring AZT were potent in inhibiting HIV-1 (Figure 20). Compounds **35** and **36** (Figure 20) were equipotent to AZT in inhibiting HIV-1 IIIB with an EC_50_ of 0.002 ± 0.001 µM. When compared with the prototypes ddI (EC_50_ of 13.3 ± 3.4 µM), E-HEPU-dM (EC_50_ of 0.007 ± 0.001 µM), and ribavirin (EC_50_ > 3.3 µM), the co-drugs **35**–**37** showed greater HIV-1 inhibitory potential. Remarkably, compared with prototype AZT (Figure 20), **35** and **36** provided greater cell protection and were significantly active against AZT and NNRTI-resistant viral strains.

While exploring the synthesis and biological evaluation of novel symmetrical nucleotide-(5′,5′)-dimer phosphotriester derivatives of AZT, McGuigan and co-workers [67] developed compounds **38** and **39** (Figure 21). Both the AZT derivatives showed anti-HIV-1 activity, and compound **39**, which has 2,2,2-trifluoroethyl, had the best inhibitory HIV-1 potential (ED_50_ of 0.4 µM) with high cell protection (CC_50_ = 600 µM). However, in JM cell assays, these compounds were less active than AZT, indicating that they may act primarily as depot forms of the free nucleoside (AZT).

Another linkage used in the design of co-drugs is the ester bond, as shown by AZT-triterpenoid hybrids **40**–**44** (Figure 22) [68]. All molecules, with the exception of **40**, showed potent anti-HIV activity. Among them, **41** was the most potent with an IC_50_ of 0.010 µM and a CC_50_ of 35 µM, which resulted in the best SI (SI = 3500) (Figure 22).

Taourirte et al. [69] used carbonates and carbamates as a linkage in co-drugs based on homo- and hetero- dimers of AZT and stavudine (d4T). The authors based their design on two arguments: 1) they expected that the linkage between the nucleosides (AZT and d4T) would not be extracellularly hydrolyzed, and the delivery and the bioavailability might be enhanced, depending on the lipophilic character of these new molecules; 2) some synergetic effects on HIV replication inhibition could be expected following intracellular hydrolysis to regenerate the two nucleosides. All carbonates were active and the homodimer **45** showed the best HIV-1 inhibition activity (EC_50_ of 0.0028 µM) (Figure 23), with a similar effect to AZT (EC_50_ of 0.0022 µM). Compound **45** was also active against HIV-2 replication, and it showed no toxicity. The opposite was observed in the carbamate series, which exhibited a weak HIV-1 inhibitory profile. This may be explained by the faster chemical and/or enzymatic hydrolytic cleavage of carbonate co-drugs leading to the active free form of AZT.

Aiming to enhance antiviral activity and improve cell membrane permeability, Matsumoto et al. [70,71] described co-drugs that combine a PI (KNI-413 or KNI-272) with AZT via an ester bond (Figure 24). As a result, compound **46** (KNI-413-AZT) (Figure 24) showed a more potent anti-HIV activity (EC_50_ of 19 nM) than AZT (EC_50_ of 126 nM) and the parent PI (EC_50_ of KNI-413 = 52 nM) [70]. Regarding conjugates KNI-272-AZT, the best results were obtained for compound **47** (EC_50_ of 0.1 nM), which features glutarylglycine as linker. Notably, **47** was 920- and 62-fold more potent than the parent PI (EC_50_ of 40 nM for KNI-272) and AZT (EC_50_ of 6.2 nM), respectively (Figure 24). This excellent result can be attributed to its better ability to cross the cell membrane and release both parent drugs inside the cell, or the better activity due to the direct interaction of the entire hybrid with its target.

Of note, while **32**, **38**–**39**, and **45** are indeed co-drugs developed with the aim of achieving a superior drug-like profile compared to the parent compound, they cannot be classified as MTDLs. This is because they share a homodimeric structure, which, following metabolic transformation, releases a single individual drug.

## 5. Conclusions

AIDS is a disease that still affects millions of people worldwide. Following the introduction of cART, its morbidity and mortality has been drastically curtailed so that today it is categorized as a manageable chronic disease and patients have a much better quality of life compared to the time when this uncommon immunodeficiency was identified in the early 1980s. In 1987, the approval of AZT, the first antiretroviral agent to tackle the disease, set a milestone in HIV/AIDS history. Since then, several other drugs and therapeutic regimens were introduced into clinics. However, side effects and viral resistance have been observed, leading to constant revisions in the therapy; the search for new and more powerful drugs with fewer side effects is imperative.

Polypharmacology has had a key role in the fight against AIDS and can still be instrumental with the development of new MTDLs [62]. A more modern alternative to drug combinations and FDCs for treating multifactorial diseases is that based on MTDLs, which can be obtained by molecular hybridization. The development of hybrids is a versatile strategy that in some applications allows the discovery of new molecules using already well-known drugs. In the AIDS drug discovery field, the design of new MTDLs based on the AZT scaffold seems highly significant.

Based on the reported examples, antiretroviral MTDLs based on AZT deserve further exploration as they may provide compounds with higher activity than the parent compound. From a chemical point of view, the structure of AZT allows for exploration in the preparation of both hybrids and co-drugs, enhancing in many cases the anti-HIV-1 profile, either by improving cell permeability (co-drugs) or through the direct interaction between the hybrids with the intended targets. However, as far as we know, since no in vivo proof-of-concept has been reported for the discussed AZT-based co-drugs and hybrids, it is impossible to predict their clinical translation and if they offer greater in vivo efficacy with respect to cART.

In summary, this review analyzes the antiretroviral activity of molecules containing the AZT scaffold, which were purposely designed to act as MTDLs. As discussed elsewhere [7], medicinal chemists should look at polypharmacology as a continuum of pharmacological opportunities, from drug combinations to MTDLs. Co-drugs and hybrids have unique features that can be effectively exploited. We argue that understanding peculiar advantages and drawbacks would be very helpful in choosing the proper anti-HIV polypharmacology strategy and in blowing the potential of MTDLs against AIDS.

## Figures and Tables

**Figure 1 molecules-27-08502-f001:**
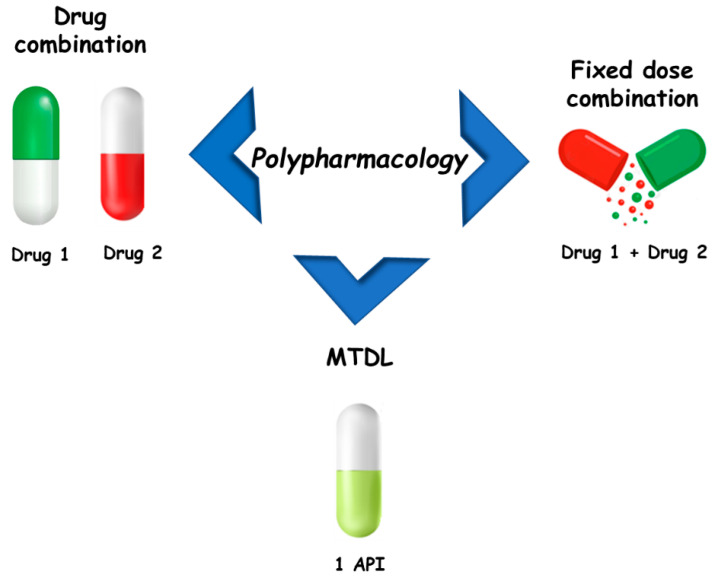
Main approaches for polypharmacology.

**Figure 2 molecules-27-08502-f002:**
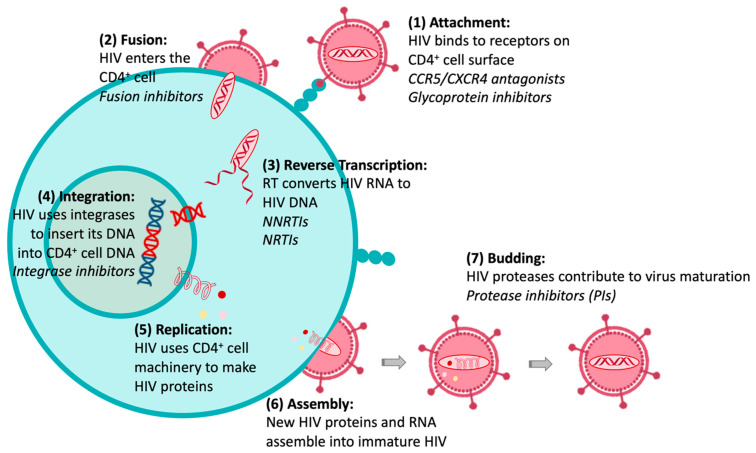
HIV replication cycle and anti-HIV therapeutic interventions (in Italics).

**Figure 3 molecules-27-08502-f003:**
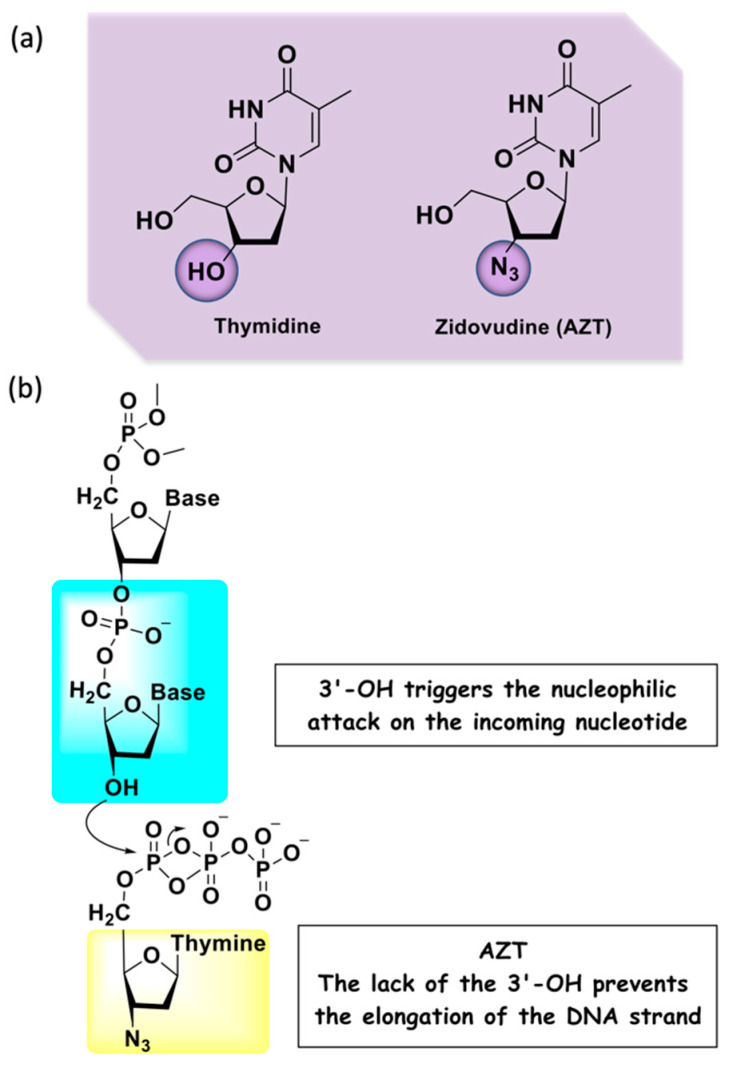
(**a**) Chemical structures of the nitrogenous base thymidine and its analog AZT; (**b**) mechanism of action of AZT.

**Figure 4 molecules-27-08502-f004:**
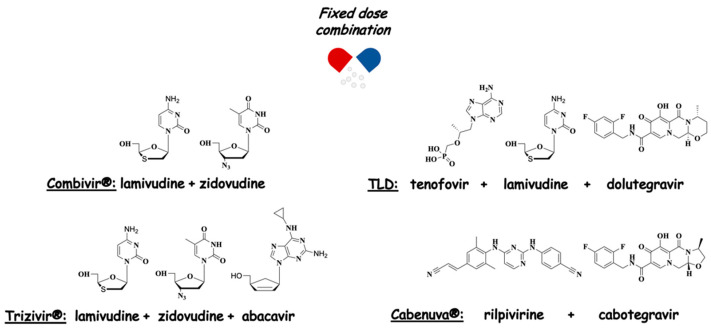
Anti-HIV FDCs.

**Figure 5 molecules-27-08502-f005:**
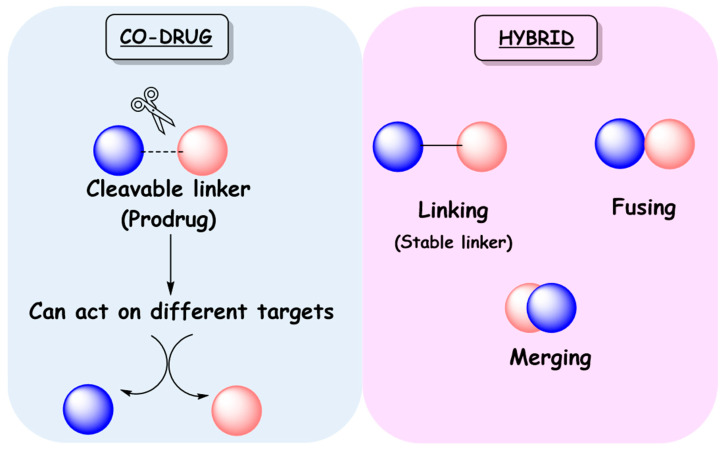
Classification of MTDLs as co-drugs and hybrids.

**Figure 6 molecules-27-08502-f006:**
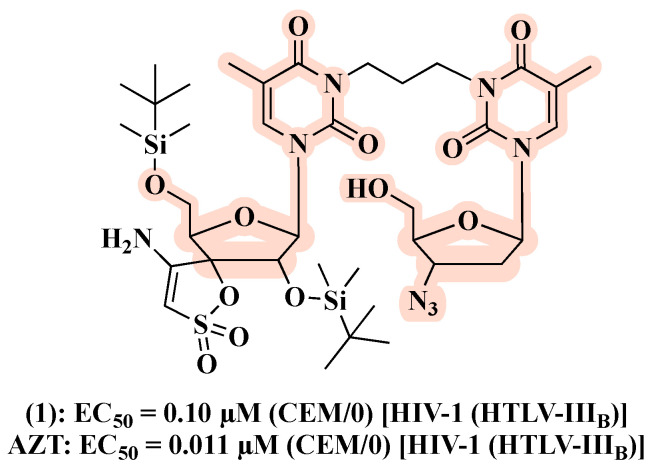
Chemical structure and antiviral activity of **1** compared to the parent compound.

**Figure 7 molecules-27-08502-f007:**
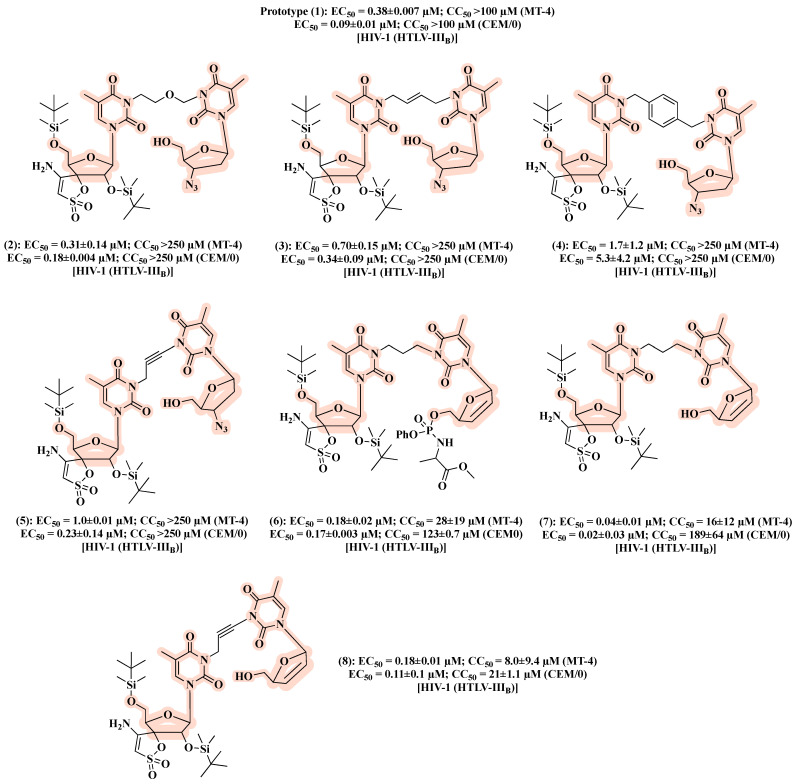
Chemical structures of **2**–**8** and their anti-HIV-1 profile.

**Figure 8 molecules-27-08502-f008:**
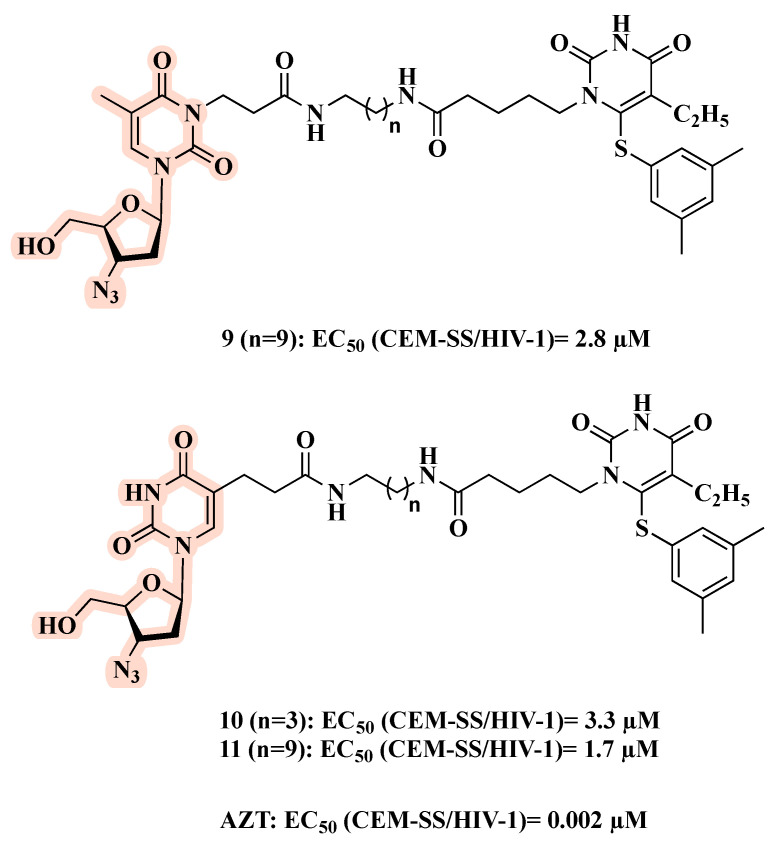
Chemical structures of **9**–**11** and their anti-HIV-1 profile.

**Figure 9 molecules-27-08502-f009:**
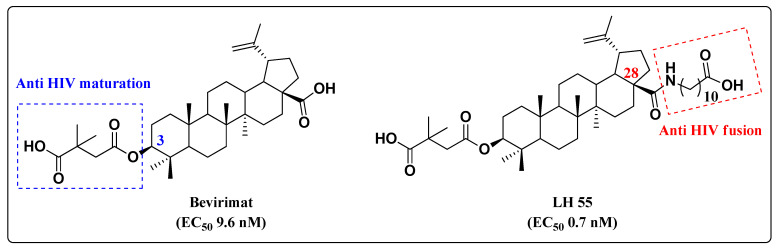
Chemical structures of betulinic acid derivatives and their anti-HIV-1 profile.

**Figure 10 molecules-27-08502-f010:**
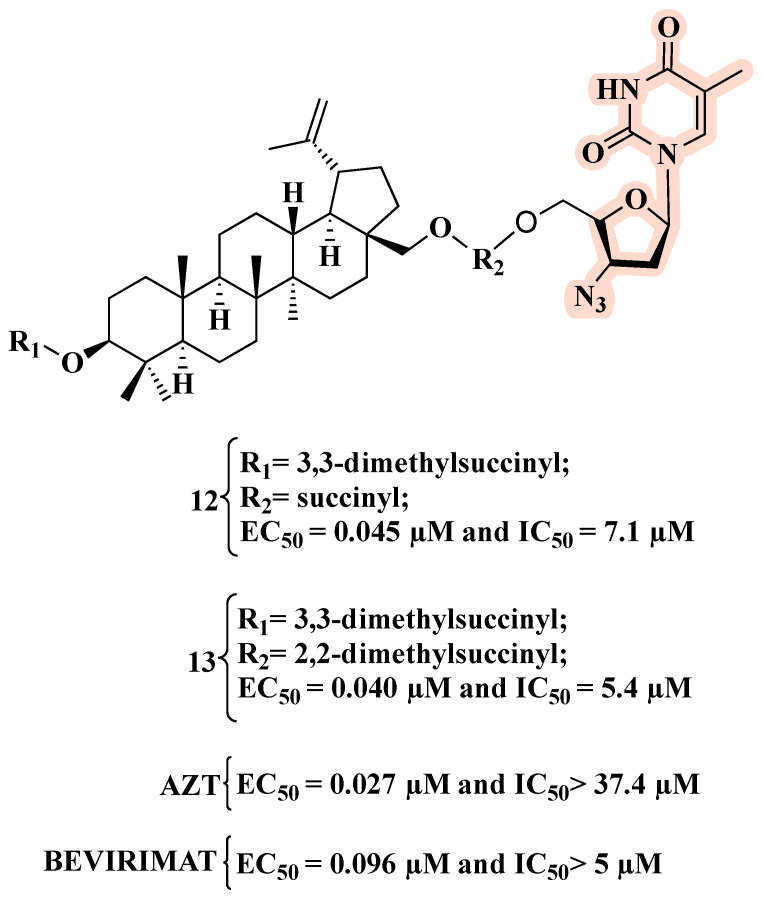
Chemical structures of **12** and **13** and their anti-HIV-1 profile.

**Figure 11 molecules-27-08502-f011:**
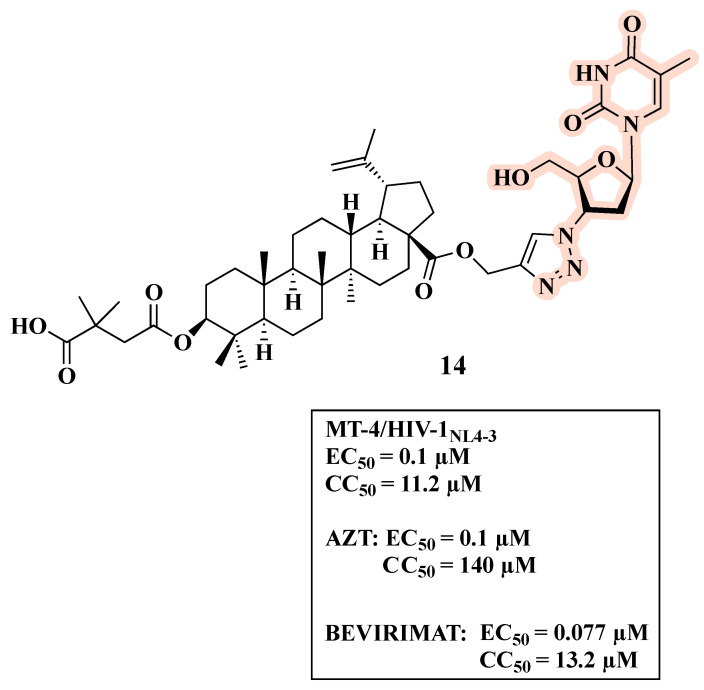
Chemical structure of **14** and its anti-HIV-1 profile.

**Figure 12 molecules-27-08502-f012:**
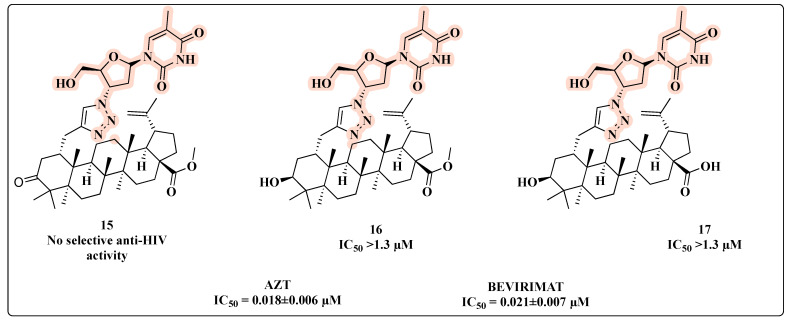
Chemical structures of **15**–**17** and their anti-HIV profile.

**Figure 13 molecules-27-08502-f013:**
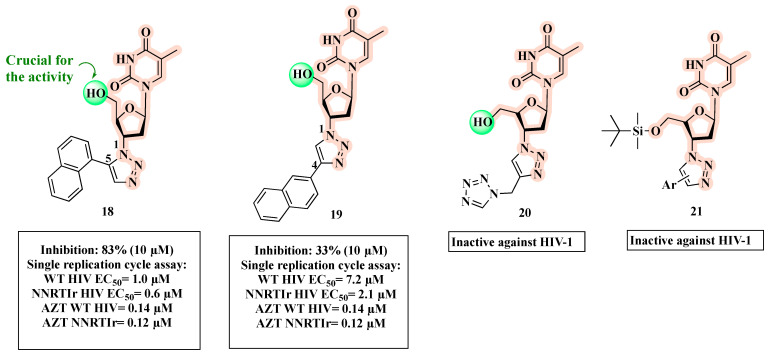
Chemical structures of **18**–**21** and their anti-HIV profile.

**Figure 14 molecules-27-08502-f014:**
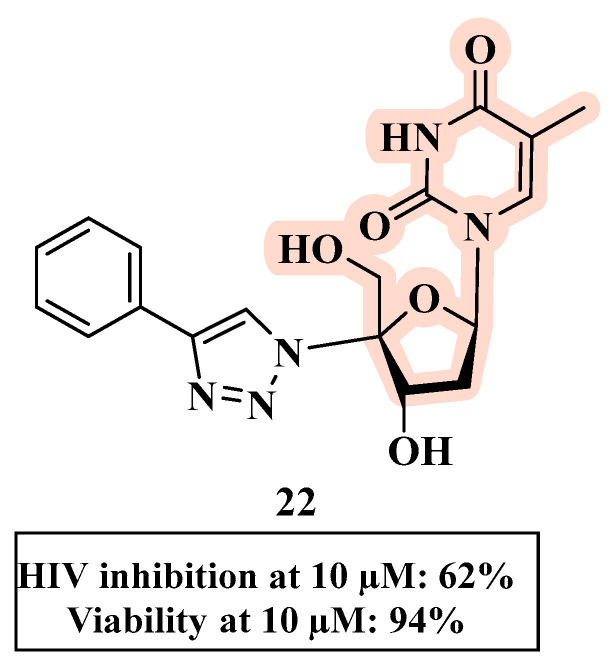
Chemical structure of **22** and its anti-HIV profile.

**Figure 15 molecules-27-08502-f015:**
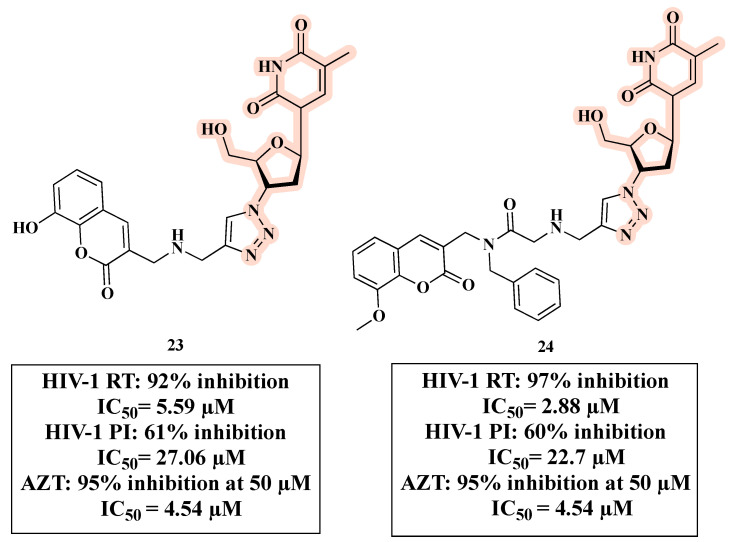
Chemical structures of **23** and **24** and their anti-HIV profile.

**Figure 16 molecules-27-08502-f016:**
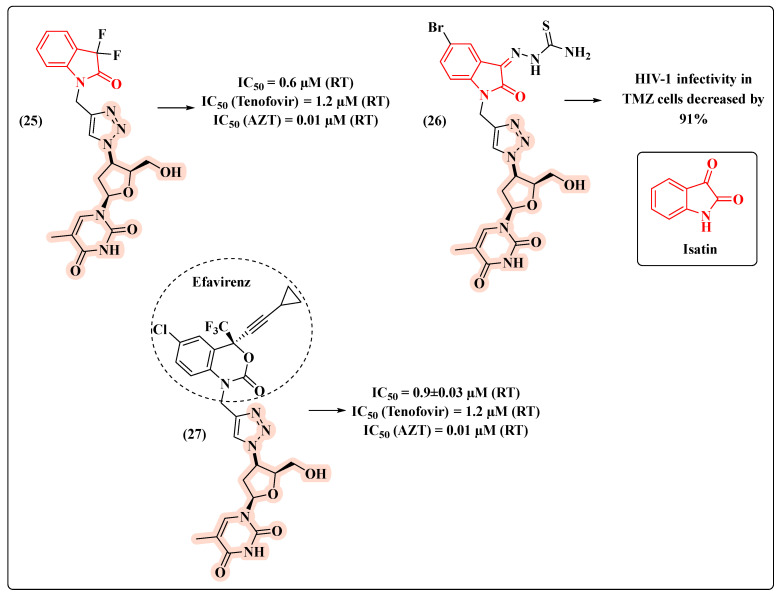
Chemical structures of **25**–**27** and their HIV-1 profile.

**Figure 17 molecules-27-08502-f017:**
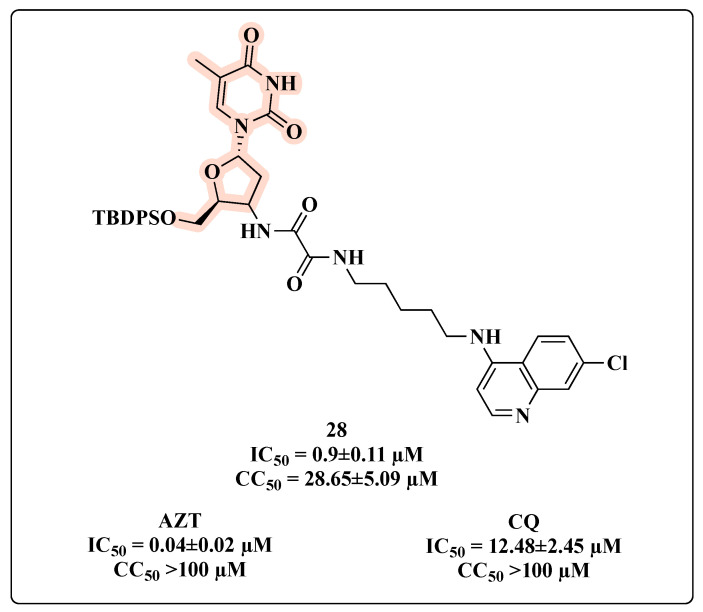
Chemical structure of **28** and its anti-HIV profile.

**Figure 18 molecules-27-08502-f018:**
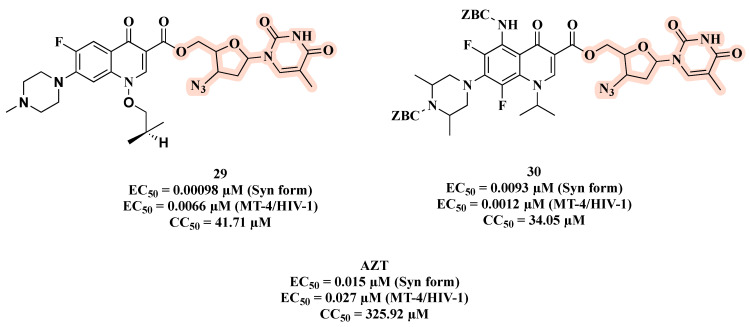
Chemical structures of **29** and **30** and their anti-HIV profile.

**Figure 19 molecules-27-08502-f019:**
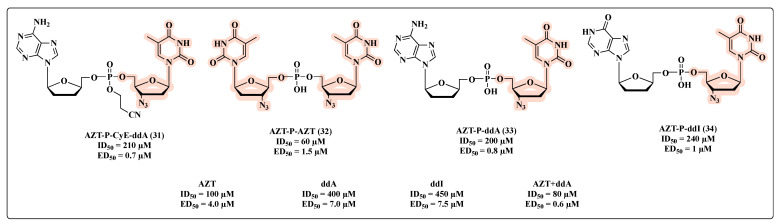
Chemical structures of **31**–**34** and their anti-HIV profile.

**Figure 20 molecules-27-08502-f020:**
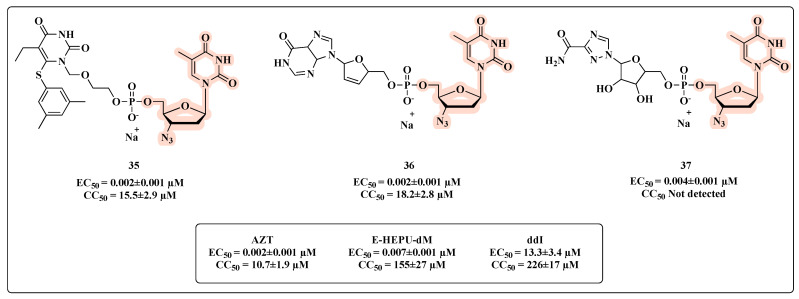
Chemical structures of **35**–**37** and their anti-HIV profile.

**Figure 21 molecules-27-08502-f021:**
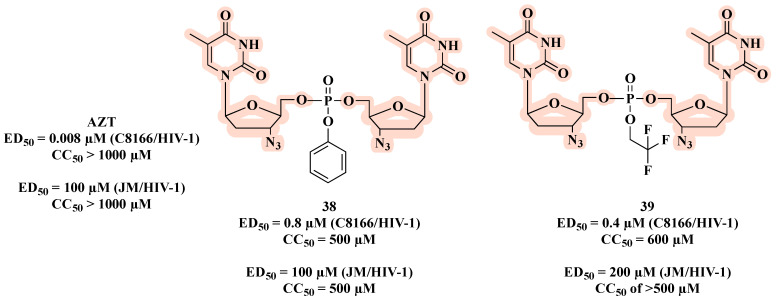
Chemical structures of **38** and **39** and their anti-HIV-1 profile.

**Figure 22 molecules-27-08502-f022:**
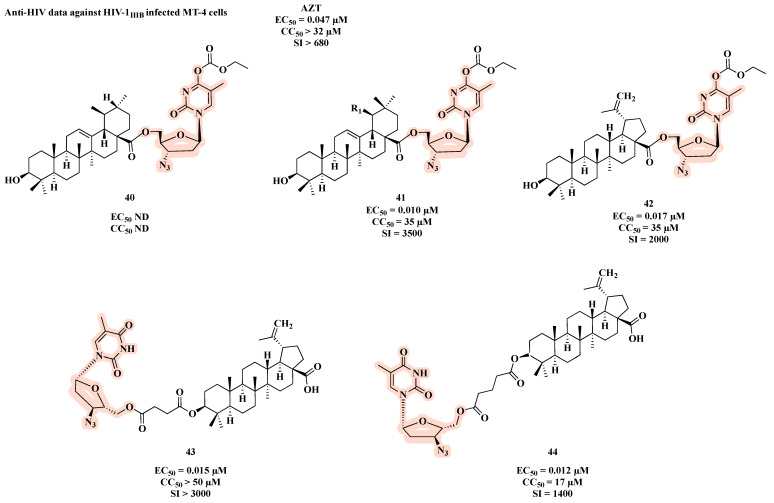
Chemical structures of **40**–**44** and their anti-HIV profile.

**Figure 23 molecules-27-08502-f023:**
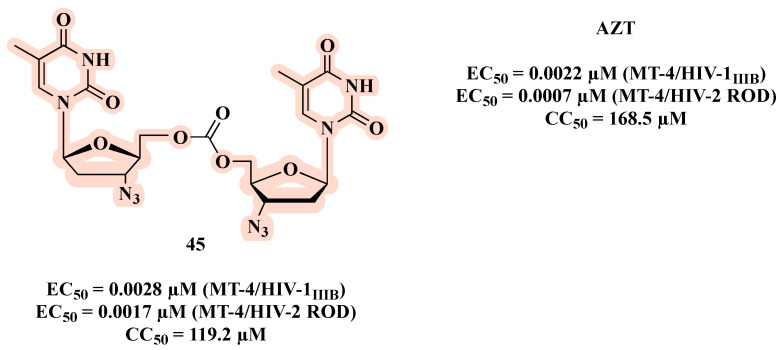
Chemical structure of **45** and its anti-HIV-1 profile.

**Figure 24 molecules-27-08502-f024:**
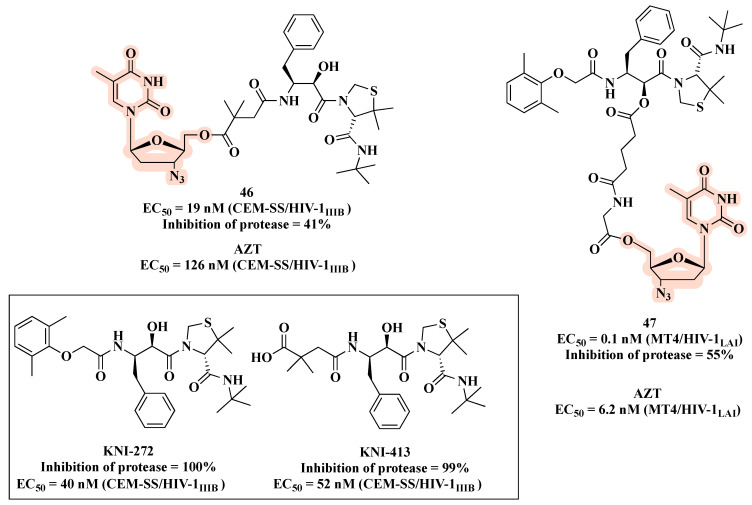
Chemical structures of PIs and co-drugs **46** and **47** along with their anti-HIV-1 profile.

## Data Availability

Not applicable.

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
