# Peer review of "The Use of Zidovudine Pharmacophore in Multi-Target-Directed Ligands for AIDS Therapy"

_molecules, 2022, doi:10.3390/molecules27238502_

Round 1

Author Response

REVIEWER #1

The aim of this review is to systematically examine and discuss AZT-based compounds as potential MTDLs active against HIV that could be useful for the treatment of AIDS.

Modulation of multiple biological targets with a single drug can lead to synergistic therapeutic effects and avoid the side effects associated with combination therapy. In addition, this strategy has several advantages over combination therapies, including more predictablepharmacokinetics, a lower likelihood of drug-drug interactions, and greater patient adherence.

The reason for developing multifunctional agents is also due to the limited efficacy and drug resistance that often occur when using single target agents, and is considered a potential therapeutic solution for complex diseases that are more likely to be cured or alleviated if multiple targets are affected simultaneously. This is essential to develop effective treatments for HIV infection, which have so far relied oncombination therapies.

This review builds on a solid foundation, analyzes an extensive and current literature, and will be of interest to a wide readership.

Nevertheless, I cannot refrain from making some critical remarks, which I hope will be carefully considered by the authors in order to clarify some points and improve the presentation of the manuscript

We would like to thank the reviewer for his/her detailed evaluation of our manuscript, for the appreciation of our work and for the suggestions to improve it. We considered all comments and we had incorporated all the suggested corrections listed below.

Compounds like 38, 39, 45 are basically prodrugs that cannot be classified as MTDL. So, are these examples consistent with the subject of this review?

We thank the reviewer for this comment. We agree with the reviewer that they cannot be classified as truly MTDLs and the text has been amended to make it clear at page 18 as follows: “Of note, while 32, 38-39, and 45 are indeed co-drugs developed with the aim of achieving a superior drug-like profile compared to the parent compound, they cannot be classified as MTDLs. This is because they share a homodimeric structure, which, following metabolic transformation, releases a single individual drug.

In addition, we have better clarified the co-drug strategy in the MTDL context at page 15 as follows: Generally speaking, the co-drug approach is based on linking two drugs through a labile covalent bond in a single molecule that acts as a prodrug, with improved therapeutic efficiency/PK/toxicity profile [7, 62]. In the specific MTDL context, the two starting molecules should be synergistic drugs, which, following metabolic transformation, have the potential to be released in the same target cells and at the same time. This is a peculiar feature of MTDL co-drugs with respect to combinations (two single drugs, each one with an individual pharmacokinetic profile) [63, 64].”

Some points of the manuscript (included in the list below) are presented too briefly and would deserve to be explained in more detail andcritically

Possible changes to be made are listed below

line n. or   

Figure

current version

suggested change

Authors’ reply

19

owing

owing to

Thanks, corrected.

110

desoxyribose

deoxyribose

Corrected.

113

bond on

binding to

Corrected.

115

can be susceptible todestruction

can be destroyed

Corrected.

120-122

Interestingly, some studies show that in some patients withadvanced infection, AZT can reverse partially the neurologicaldysfunctionsassociated with HIV,such as dementia andperipheral neuropathy

Rephrasing this sentence, e.g. Interestingly, somestudies show that AZT can partially reverse HIV-associated neurologicdisorders, such as dementia and peripheral neuropathy,in some

patients with advancedinfection.

Corrected.

122

reduced

Diminish

Corrected.

127

headaches

Headache

Corrected, thank you.

Figure 4

the color coding of the drugs is a bit confusing. I suggest labeling lamivudine and zidovudine green and red, respectively, and assigningabacavir the color blue.

Thank you for pointing this out. According to the reviewer’s suggestion, we have modified Figure 4.

Figure 4

Is 2 API’s correct? I think itshould be 2 APIs

Corrected.

163

may have betterpatient tolerance

may be better tolerated bypatients

Corrected.

167-168

Whereas hybrid molecules can becomposed of two or morepharmacophores with establishedactivity and/ortoxicity, united in asingle compound

Rephrasing this sentence, e.g. Conversely, hybrid molecules may consist of two or more pharmacophores with proven activity and/or toxicity combined in a singlecompound

Corrected.

184

when two targets belonging to the same biochemical pathways they areclassified

Delete when and they

Corrected.

187-189

To perform the rational design of a MTDL is necessary to have a broadtheoretical base, andemploys classical concepts of medicinal chemistry, likemolecularhybridization, whichis   

one of the main toolsused.

I suggest rephrasingthis sentence,

e.g. A broad theoretical basis is required for the rational design of an MTDL. Classical concepts ofmedicinal chemistry areused, such as molecular hybridization, which is one of the most important tools.

Thanks, corrected.

203

an half maximal

a half-maximal

Corrected.

206

toward decreasingpotency

to decreasing potency

Corrected.

236-237

Nevertheless, synergistic effectswere not observed,maybe because the linker is unable to permit the AZT andthe HEPT motifs tobind simultaneously at their respective  sites

Moreover, no synergistic effects were observed, perhaps because the linker is unable to allowsimultaneous binding of theAZT and HEPT motifs at theirrespective sites

Corrected.

256-257

Based on this and inthe search for

new…

On this basis, and in searchof

new…

Corrected.

259-260

with that

to those

Corrected.

276

is equal

Corresponds

Corrected.

Figure 12

Both 16 and 17 show IC50 =CC50: Would you comment? In lines 284-286 of the text you only wrote: …whereas 16 and 17 exhibited a certain degree of HIVinhibitory activity but lower than AZT (Figure 12).Cytotoxicity was also evaluated using the MTTassay and 15-17 were safer than the prototype AZT.

We thank the reviewer for raising this point. We went back to the original article and, as the authors reported that “none of the three hybrids conjugated with AZT (15, 16 and 17) displayed significant anti-HIV activity”, we consider not appropriate to discuss about toxicity and CC50 values. According to her/his suggestion, we have now modified the text as follows: “However, none of the three hybrids conjugated with AZT (15, 16 and 17) displayed significant anti-HIV activity (Figure 12).”

Figure 13

Was the data in the box forcompounds 18 and 19 givencorrectly? In any case, this data is cryptic and it would be better to explain it inmore detail in the text

We thank again the reviewer for pointing this out and we apologize for not being clear. We checked the data for compounds 18 and 19 and they were reported correctly in the figure. We have now explained it in more detail as follows: “However, 18 and 19 showed a reduced anti-viral activity in a single replication cycle WT HIV assay (EC50 = 1 µM and 7.2 µM, respectively) compared to AZT (EC50 = 0.14 µM). The authors further characterized the antiviral profile of 18 and 19 against a NNRTI-resistant (NNRTIr) HIV strain (Figure 13). Notably, 18 (EC50 = 0.6 µM) had 3.5-fold higher efficacy than 19 (EC50= 2.1 µM) against the resistant strain, but 5-fold lower than AZT (EC50 = 0.12 µM).

303-304

instead of at

rather than at

Corrected.

304-305

In spite of presenting the best anti- HIV profile within the series, 22 had amoderate anti-HIV-1

activity…

Although 22 had the bestanti-HIV profile within the series, it had moderate anti-HIV-1 activity…

Corrected.

313

fashion, resulting asdual-acting…

manner as dual-acting…

Corrected.

Figure 17

Is the structure of compound 28 correct? I think 28 should have 5methylene groups in the linker, please check. Moreover, the C-5’ OH is protected: no comment?

We thank the reviewer for spotting this. The structure of compound 28 has been checked and changed accordingly in Fig 17. In addition, we have now commented on the protected C-5’ OH, as follows: “Taking into consideration only the HIV inhibitory activity, compound 28, featuring a protected C-5’ OH-AZT linked to CQ thorough a succinyl spacer, resulted the most potent antiviral hybrid with an IC50 of 0.9 µM (Figure 17). This was in part expected since additive in vitro anti-HIV effects were observed in AZT-CQ combination. The presence of a protecting group on the hydroxyl function seemed important for anti-HIV effects. Nevertheless, 28 displayed a reduced activity compared to AZT (IC50= 0.04 µM), but greater than that of CQ (IC50 = 12.48 µM).”

353

…a SI of 148.9.

Is it true?

We thank again the reviewer for this comment. We re-calculated the SI of compound 29 and it corresponds to a value >6000. Accordingly, we corrected the SI value in the text.

Figure 19

Compound 32 is base on two AZT units.

Please, check the structureof compound 32

Thank you for spotting this, we have now corrected the structure of compound 32 in Figure 19.

390

on inhibition

in inhibiting

Corrected.

393-395

Remarkably, 35 and36 provided major cell

protection when compared to theprototype AZT (Figure 20) and were significantly active against AZT and NNRTI-resistant virusstrains.

Remarkably, compared withprototype AZT (Figure 20), 35 and 36 provided greater cell protection and were significantly active against AZT- and NNRTI-resistant  viral strains.

Corrected.

455

unusual immunedeficiency

uncommonimmunodeficiency

Corrected.

478-480

We argue that understanding theiradvantages and drawbacks is veryhelpful in choosing a proper and fully blow the potential of

winning the fightagainst AIDS.

This sentence is not clear;please   

rephrase.

We apologize for not being clear. We have now rephrased the sentence as follows “We argue that understanding peculiar advantages and drawbacks would be very helpful in choosing the proper anti-HIV polypharmacology strategy and in blowing the potential of MTDLs against AIDS.

Once the manuscript is properly revised, I strongly recommend publishing it on Molecules

Reviewer 2 Report

It would be noteworthy to indicate in the introduction some differences between multitarget and multifunctional drugs based on a nice paper presented by Kleczkowska (2022; doi: 10.3390/ijms23073739).

Please try to provide potential mechanisms of drugs activity against HIV and AIDS by presenting figure.

Since it is written that most of the drugs should be administered simultaneously with other antiviral drugs the paper would be greatly improved if the Authors introduce some detailed characteristic of HIV virus itself (in terms of its mechanism, resistance, etc). Then, the reason to take at least 2 different drugs will be given. 

The Authors should state the type of study when demonstrating every hybrids activity  (EC50). In line with this, are there any papers in the literature presenting antiHIV activity of hybrid vs. its pharmacophores (single or administered as a mixture)? If yes, it would be good to provide some information, which additionally may confirmed hybrids great activity.

The Authors used sometimes EC50 and IC50. It would be nice to unify the values. 

Since, chimeric drugs are designed in order to reduce possible side effects that may be generated by a popular polytherapy, my question is whether there are any studies in vivo demonstrating the safety profile of such drugs?presented?

Author Response

REVIEWER #2

It would be noteworthy to indicate in the introduction some differences between multitarget and multifunctional drugs based on a nice paper presented by Kleczkowska (2022; doi: 10.3390/ijms23073739).

We apologize with the reviewer, but we could not find a definition of multifunctional drugs in the suggested paper.  In agreement with this comment that there are several definitions of multi-target compounds, but we have decided to use the definition of MTDLs, which is one of the most-accepted one in the literature, as explicitly reported at page 2: “Regarding multi-target drugs, in 2008 Bolognesi and coworkers, based on the potential of polypharmacology in the treatment of neurodegenerative diseases, proposed the term multi-target-directed ligands (MTDLs) to refer to this class of compounds [6, 7] This was done with the purpose of better highlighting “their ability to interact with the multiple targets thought to be responsible for the disease pathogenesis” and clearly differentiate them from so-called “promiscuous drugs” [1]. On this basis, in this review the term MTDLs will be used to refer to these compounds”.

Please try to provide potential mechanisms of drugs activity against HIV and AIDS by presenting figure. Since it is written that most of the drugs should be administered simultaneously with other antiviral drugs the paper would be greatly improved if the Authors introduce some detailed characteristic of HIV virus itself (in terms of its mechanism, resistance, etc). Then, the reason to take at least 2 different drugs will be given. 

We thank the reviewer for highlighting this point. According to her/his suggestion, we have now made a new Figure (see Figure 2), reporting the HIV replication cycle along with the possible therapeutic interventions. We have also described HIV virus in terms of mechanism of replication and resistance at pages 2-3, as follows: “Figure 2 illustrates that the HIV life cycle starts when (1) HIV fuses with the CD4 cell membrane, and (2) a capsid consisting of the virus’s genome and proteins moves into the cell. (3) The disruption of the capsid shell allows HIV reverse transcriptase to transcribe the viral RNA into DNA. Then, (4) HIV DNA is transported across the nucleus, where the HIV integrase inserts the HIV DNA into the CD4 cell DNA. (5) The host transcription machinery transcribes HIV DNA into new RNA copies, which will be used for the genome of a new virus and to make new HIV proteins. (6) The new HIV RNA and proteins move to the cell surface, where an immature HIV forms. Finally, (7) the virus is released from the cell, and HIV proteins called proteases cleave newly synthesized proteins to create a mature infectious virus. The available drugs act at different stages of HIV replication cycle and through different mechanisms of action [11]: (i) as inhibitors of the reverse transcriptase (RT) enzyme (RTIs), either nucleosides (NRTI) or non-nucleoside (NNRTI); (ii) by blocking the fixation or fusion process, collectively called "entry inhibitors"; (iii) acting on glycoproteins (gp120 and gp41) or receptors and co-receptors present on the host cell surface (CD4 and CCR5 or CXCR4); (iv) as inhibitors of integrase, the enzyme that promotes active integration of the viral DNA double-strand into the host cell genome (integrase inhibitors), II); or by "competitive" inhibition of proteases (protease inhibitors, PI). However, drug resistance may arise from each of these drug classes. There are essentially two mechanisms by which resistance to NRTI can occur: (i) mutations of residues at or near the drug-binding site, and (ii) mutation of the residues that results in reduced incorporation/enhanced removal of the drug into/from its binding site. For NNRTI and PI drug classes, the resistance occurs primarily as a result of amino acid mutations within or proximal to the drug-binding site. Drug combinations have been shown to slow down the evolution of resistance, as the simultaneous administration of two drugs, which operate with different mechanisms of action, can reduce the probability of mutations.”

The Authors should state the type of study when demonstrating every hybrids activity (EC50). In line with this, are there any papers in the literature presenting antiHIV activity of hybrid vs. its pharmacophores (single or administered as a mixture)? If yes, it would be good to provide some information, which additionally may confirmed hybrids great activity.

We are thankful to the reviewer for highlighting this. Where this information was not already reported, we have now added it throughout the text and accordingly we have updated the corresponding figures:

Page 7 and Figure 8: “However, 9-11 were less active than AZT (EC50 = 0.002 µM).”

Page 8 and Figure 10: “Particularly, 12 and 13 (Figure 10) displayed EC50 values against HIV-1NL4-3 infected MT-4 cells comparable to those of AZT and bevirimat. Importantly, 12 and 13 outperformed both parent compounds in terms of inhibition of mock-infected MT-4 cell growth (IC50).”

Pages 9-10 and Figure 11: “As an example, compound 14 (Figure 11), featuring a 1,2,3-triazole as a spacer between AZT and the C-28 position of betulinic acid, exhibited an EC50 value of 0.10 µM, which corresponds to that of AZT (EC50 = 0.10 µM) and is higher than that of bevirimat (EC50 = 0.077 µM) [37]. Compound 14 also displayed toxicity comparable to bevirimat (CC50 of 11.2 and 13.2 µM, respectively), but greater than AZT.”

Page 12 has been updated with the EC50 values of parent compounds.

Page 14 and Figure 17: “Nevertheless, 28 displayed a reduced activity compared to AZT (IC50 = 0.04 µM), but greater than that of CQ (IC50= 12.48 µM).”

Figure 24 has been updated with the EC50 values of parent compounds. They have been already commented in the text.

The Authors used sometimes EC50 and IC50. It would be nice to unify the values. 

We thank the reviewer for this comment. However, we discussed activity data as reported in the original research articles.

Since, chimeric drugs are designed in order to reduce possible side effects that may be generated by a popular polytherapy, my question is whether there are any studies in vivo demonstrating the safety profile of such drugs?presented?

Thank you for pointing this out. To the best of our knowledge, none of the discussed AZT-based MTDLs have been tested in vivo. We have now added a sentence in the Conclusions as follows: “However, as far as we know, since no in vivo proof-of-concept has been reported for the discussed AZT-based co-drugs and hybrids, it is impossible to predict their clinical translation and if they offer greater in vivo efficacy with respect to cART.”

Round 2

Reviewer 1 Report

I apologize for not pointing out an error in Figure 10 when I first revised the manuscript. There is a missing oxygen atom to link AZT to the succinyl chain: It should be -O-R2-O-.

Please change "thorough" to "through" (line 371).

With these corrections I consider the manuscript publishable on Molecules.

Author Response

I apologize for not pointing out an error in Figure 10 when I first revised the manuscript. There is a missing oxygen atom to link AZT to the succinyl chain: It should be -O-R2-O-.

We thank the reviewer for spotting this. Accordingly, the structure reported in Figure 10 has now been corrected.

Please change "thorough" to "through" (line 371).

Thanks, corrected.

With these corrections I consider the manuscript publishable on Molecules.

Reviewer 2 Report

The Authors have greatly improved the manuscript which is now suitable for publication. Thank you and good luck.

Author Response

The Authors have greatly improved the manuscript which is now suitable for publication. Thank you and good luck.

We thank the reviewer for his/her positive feedback.